# “A Judgment-Free Zone”: Adaptation and Pilot Study of a Virtual Wellness Group for African American Mothers with Young Children

**DOI:** 10.3390/ijerph21040390

**Published:** 2024-03-23

**Authors:** Kimberly M. Brooks, Dominique Charlot-Swilley, Hillary A. Robertson, Nia Bodrick, Aimee L. Danielson, Marta Genovez, Claire Boogaard, Sydney Morris, Sanyukta Deshmukh, Lauren Kiker, Olukemi Green, Huynh-Nhu Le

**Affiliations:** 1Children’s National Hospital, 1 Inventa Place 5th Floor, Silver Spring, MD 20910, USA; nbodrick@childrensnational.org (N.B.); mgenovez@childrensnational.org (M.G.); cboogaar@hschealth.org (C.B.); srdeshmukh@som.umaryland.edu (S.D.); 2Department of Psychiatry, Medstar Georgetown University Hospital, Washington, DC 20007, USA; dc1203@georgetown.edu (D.C.-S.); hillary.robertson@georgetown.edu (H.A.R.); aimee.l.danielson@gunet.georgetown.edu (A.L.D.); 3Department of Psychology, Palo Alto University, Palo Alto, CA 94308, USA; smorris@paloaltou.edu; 4Department of Psychological and Brain Sciences, George Washington University, Washington, DC 20013, USA; lmkiker@alaska.edu (L.K.); ogreen92@gwmail.gwu.edu (O.G.); hnle@gwu.edu (H.-N.L.)

**Keywords:** perinatal depression/anxiety, prevention, African Americans, mixed methods, pediatric

## Abstract

The COVID-19 pandemic has been particularly challenging for the mental health of African American (AA) birthing people. The pandemic necessitated shifting mental health care to online interventions. The goals of this study were to (1) describe an adapted evidence-based group preventive intervention for AA mothers with young children within a pediatric setting and (2) evaluate the feasibility, acceptability, and preliminary effectiveness of this virtual intervention. Phase 1 describes the adaptation of the HealthySteps Mom’s Virtual Wellness Group, including eight weekly sessions based on the Mothers and Babies Course. Phase 2 was a mixed-methods, pre–post intervention design. Six AA mothers with young children completed questionnaires related to depression, anxiety, and parenting competence at three time points: pre-intervention (T1), post-intervention (T2), and 3 months post-intervention (T3). The participants also completed a focus group post-T2 to gather qualitative feedback regarding the intervention. The median scores for depression were lower at T2 and increased at T3, and for anxiety, they increased at T2 and decreased at T3. The median scores for parenting competence increased across the three time points. The participants attended a mean of 7.2 sessions (SD = 0.74). The qualitative results indicate that the participants gained a sense of empowerment, enjoyed connecting with other mothers, and acquired information. This pilot study suggests that a virtual intervention is feasible, acceptable, and can increase parenting competence and support among AA mothers with young children.

## 1. Introduction

Perinatal mood and anxiety disorders (PMADs) are a group of mental health disorders that affect birthing people during pregnancy and postpartum periods. Approximately 15% to 21% of birthing people experience PMADs [1,2,3]. Birthing individuals diagnosed with PMADs may experience a range of symptoms, including depression and anxiety, as well as changes in their functioning and relationships with those around them. When left untreated, PMADs can result in negative consequences for birthing people and their families [4]. Therefore, the early identification, prevention, and treatment of PMADs are crucial.

PMADs disproportionately affect particular subgroups of birthing people. Compared to the general population, Black and Latinx birthing people experience PMADs at higher rates [5]. Black birthing people experienced more symptoms and greater severity of postpartum depression (PPD) even when other risk factors were controlled [6]. In particular, the impact of generational racism on the bodies of Black birthing individuals is a significant factor in maternal and birth outcomes [7,8,9].

The COVID-19 pandemic has been a unique challenge for birthing people, with a demonstrated increase in rates of PMADs [10,11]. Restricted social interactions, a lack of social support, increased isolation and loneliness, a shift in maternity and perinatal care, and fear of life’s impermanence all likely contributed to an increased incidence in PMADs [12]. In addition, some researchers have noted that historical inequities were exacerbated by COVID-19 [13]. There are many reasons for the disparity in PMADs among Black people versus other groups. In addition to dealing with the depression and anxiety that accompanies giving birth, AA women also face structural and interpersonal racism and potential stressors such as residing in a more disadvantaged neighborhood [14], distrust of the medical system, and a lack of culturally sensitive care [15]. For example, in a study of 151 Black birthing individuals, these experiences of interpersonal racism, structural racism, and negative COVID-19 pandemic experiences were associated with greater risk for postpartum depression and anxiety [16]. The added stress of the pandemic on an already overburdened African American community has been documented [16]. Given these mental health concerns and the structural disparities in treatment, more attention towards effective interventions to address PMADs, particularly for birthing people of color, is warranted.

Despite the challenges associated with the pandemic, the pandemic also offered an opportunity to innovate ways to provide obstetrics and mental health care to birthing people [17,18]. However, during the pandemic, there was limited attention given to how mental health care could be provided in pediatric settings, where historically the focus has been on the baby and child. Acknowledging that parental mental health affects infant and child physical and mental health, HealthySteps, an evidence-based national model, integrates a child development specialist into pediatric primary care [19]. Pediatric primary care is the ideal setting for the prevention of PMADs during the first few years of life, with parent–child touchpoints during well-child visits. In Washington, DC, an enhanced version of HealthySteps (HealthySteps DC) is currently being implemented that includes a perinatal mental health provider that provides PMADs screening and brief mental health interventions for parents with positive screens. Due to COVID-19, these services were limited to brief touchpoints during well-child visits, leaving the parents (most often mothers) socially isolated and at risk for PMADs during the birthing period. Thus, we developed an online intervention intended to decrease risk for PMADs and increase support during this period for parents in HealthySteps DC. The aims in this study were twofold: (1) to describe the adaptations made to an evidence-based group preventive intervention for African American (AA) mothers with young children within a pediatric setting and (2) to present the results from a mixed-method pilot study to examine the feasibility and acceptability of the adapted intervention for AA mothers in the midst of the COVID-19 pandemic.

## 2. Materials and Methods

### 2.1. Phase 1

#### The HealthySteps Moms Virtual Wellness Group: Intervention Development

This was a group intervention of eight weekly sessions, which were based on the psychoeducation and prevention models derived from the Mothers and Babies Course, the MBC [20,21]. The MBC is an evidence-based preventive intervention for perinatal depression based on cognitive–behavioral and attachment-based theoretical frameworks originally developed for low-resourced English- and Spanish-speaking women [20,22]. This intervention has been adapted for multiple populations, settings, and countries [21].

In this study, the MBC kept its eight-session format and was adapted in several ways; this process occurred over a 6 month period in 2021, reviewing and revising the intervention with input from all authors [23]. First, we expanded the MBC to address not only depression but also anxiety and parenting stress for AA mothers with young children. Second, in each session, we expanded the content to integrate mindfulness practices to manage distress. The mindfulness practices varied in nature and content, with the overall aim to bring awareness to sensations, thoughts, emotions, and the surrounding environment. By cultivating self-awareness, mothers can better understand their own experiences and responses, which can facilitate more mindful and intentional moments of connection with their babies. The specific practices are listed in Table 1. Third, we considered and integrated culturally specific content to reflect the experiences of AA mothers with young children (e.g., affirming identity as a Black woman and mother in session 2). Fourth, we also included more local resources for AA parents in the DC area. Relatedly, we included a guest speaker in session 4, a reproductive psychiatrist, to address psychiatric medication issues and concerns, and in session 7, a pediatrician or developmental psychologist, to address any questions that the participants had about infants’ developmental milestones. The guest speakers provided a general overview of possible issues that can arise during the perinatal period and allowed time for the mothers to ask their own questions. These discussions were then integrated into the content of that particular session. For example, session 4 addressed the relationship between mood and thoughts; thus, session 4 inquired more about the participants’ thoughts regarding how medication can help to improve mood. These changes resulted in a participants’ and facilitators’ manual called the Healthy Steps Moms Virtual Wellness Group [23].

Each group session lasted for 90 min, delivered virtually via the Zoom platform (Table 1). The interventions were structured conversations around chosen topics relevant to parental mental health with skills-building techniques facilitated by two clinicians, a clinical psychologist and a licensed professional counselor, with maternal mental health and early childhood expertise. The first facilitator (and first author) is a clinical psychologist with graduate and post-graduate training in maternal mental health and infant mental health, and the second facilitator is a licensed professional counselor who has worked in private and public settings for over 10 years supporting mothers and families with young children. Every session began with a brief mindfulness exercise, the majority of which were led by a psychologist trained in mindfulness and maternal mental health. In session 4, the group included a guest speaker (a reproductive psychiatrist, to answer questions about medication management during the perinatal period). Another innovation in the adapted version is the integration of texts in between sessions. Text reminders between sessions were sent to increase the utilization of skills, coordinate logistics, and set homework reminders. Following the completion of each session, four texts were sent to the participants throughout the week on four topics as related to that session’s content: (a) skills reinforcement (e.g., in the thoughts module, “Have you noticed any unhelpful thoughts you have? Reply Yes/No”); (b) homework reminders (e.g., “Keep track of your mood, notice how many helpful/unhelpful thoughts you have”); (c) mindfulness content (e.g., provide links to practice at home); and (d) self-monitoring of mood (ratings 1–9, 1 = lowest, 9 = highest). In addition, texts were sent regarding logistics (e.g., session reminders, coordinating meal distribution). All the participants received these text messages between sessions on a bidirectional text messaging platform known as “HealthySMS”.

### 2.2. Phase 2

#### 2.2.1. Study Design

This study had a mixed-methods, pre–post intervention design, without a control group. This study was reviewed and approved by the Institutional Review Boards at the universities.

#### 2.2.2. Study Site

HealthySteps DC, through support from the Early Childhood Innovation Network (ECIN), has been part of two community-based primary care centers in Washington, DC, since November 2016. The ECIN is a local collaboration of health and education providers, community-based organizations, researchers, and advocates committed to catalyzing system change for child- and family-serving entities in Washington, DC.

The national HealthySteps program involves the following core components: (1) team-based well-child visits conducted jointly between pediatricians and the HealthySteps Specialist involving child development guidance, parent coaching, and the dissemination of early learning resources; (2) screening that includes an assessment of child development, social–emotional skills, and behavioral functioning in addition to family protective/risk factors and social determinants of health; (3) connection to community resources; and (4) access to mental health support between well-child visits for families with a greater need for support [17]. HealthySteps is a tiered model where all children aged 0–3 receiving pediatric primary care receive Tier 1, including components 1 and 2, which are considered universal services. Families receiving community resource referral facilitation and/or short-term consultation on child development or parent coaching are considered Tier 2. Families who elect to access ongoing mental health support from a HealthySteps specialist are considered Tier 3 and will be assessed at all well-child visits.

#### 2.2.3. Participants and Procedures

The participants were recruited from August to September 2021 from two pediatric primary care sites. The two pediatric primary care sites are located in Wards 7 and 8 in Washington, DC. Both wards are located in large urban communities that have been historically impacted by structural and systemic inequities. The population served is 90% non Hispanic Black and 3% Hispanic. The majority of the population served is Medicaid-insured (77%). These two clinics serve several thousand patients a year. The inclusion criteria included (a) primary female caregivers with a legal right to register their child for services at the clinic; (b) children aged 0–3; and (c) being eligible for the HealthySteps DC program per determination from HealthySteps staff. The exclusion criteria included (a) active, untreated substance abuse; (b) current self-harm ideation; and (c) active involvement with Child Protective Services. The eligible participants were referred to the virtual group by health providers at the site, including pediatricians, social workers, Family Service Coordinators (FSCs) and HealthySteps Specialists. In particular, the FSCs who had the most contact with the participants called the eligible participants to ask if they were interested in participating in the study. If participants were interested, the FSCs contacted the research team to follow up with providing additional study information and obtaining informed consent. The participants were compensated with gift cards for completing questionnaires at three time points (Time 1 = USD 30, T2 = USD 30, T3 = USD 40) and taking part in the focus group (T4 = USD 40) following completion of the post-intervention. The participants were also provided with resources, such as diapers, snacks, and gift cards for meals, which occurred every 3 times during the 8 sessions, to increase intervention attendance. Once consent was obtained, one of the group facilitators contacted all of the participants to do a pre-orientation meeting (Pre-T1), providing information about the group and assessing social needs that may require referrals prior to orientation.

A total of 11 mothers were approached about the study. Of these, five did not participate: one was not interested, one had a time conflict with the intervention, and three could not be contacted for follow-up. The remaining six mothers consented and participated in the study (Table 2). The participants were AA caregivers with an average age of 29.3 years (SD = 6.0). The majority of the participants were single (83.3%, n = 5) but parenting with a partner (66.7%, n = 4). Two-thirds of the participants were first-time mothers (66.7%, n = 4). Education level varied across the group: one-third of the participants had a high school diploma or GED (33.3%, n = 2), one-third had completed trade/vocational school (33.3%, n = 2), and one-third had a graduate degree (33.3%, n = 2). On average, the participants attended 7.2 sessions (SD = 0.74), and all the participants completed the intervention (100%, n = 6).

#### 2.2.4. Data Collection and Analysis

Given the small sample size, the results from the quantitative data did not yield any significant statistical value and should be interpreted with caution. Instead, we emphasize the results from the qualitative data from the focus group, as described below.

##### Quantitative

Quantitative data, via the self-report measures described below, were collected at three time points. Time 1 (T1) and Time 2 (T2) took place pre- and post-intervention, respectively (August and November 2021), and Time 3 (T3) took place three months following the completion of the intervention (February 2022). Data were collected via Qualtrics, an online survey platform. All quantitative data were cleaned, managed, and analyzed using IBM SPSS 29.0. Descriptive statistics were calculated. In addition, data from text messages were analyzed based on skill reinforcement, homework reminders, mindfulness content, self-monitoring of mood, and logistics.

##### Measures

The participants were asked to complete a demographic questionnaire that collected information about their race, gender, age, marital status, education level, employment status, income, household size, number of children, and age of their youngest child. Six items adapted from the Coronavirus Impacts Questionnaire [24] were used to assess the participants’ experiences with COVID-19. It utilized a 7-point scale to rank the degree to which the participants agreed with the statements (1, *not true for me at all*, to 7, *very true for me*). Higher scores indicated that the pandemic had a greater impact, with scores ranging from 7 to 42.

The Edinburgh Postnatal Depression Scale (EPDS) [25] was used to assess postpartum depressive symptoms within the past 7 days. The 10-item questionnaire is scored on a 4-point scale to assess symptom severity, with higher scores indicating a higher risk for depression.

The General Anxiety Disorder Questionnaire-7 (GAD-7) [26] was used to assess levels of anxiety during the prior two weeks. The 7 items utilize a 4-point scale to rank symptom severity. Scores are totaled with a possible range from 0 to 21. Scores ranging from 5 to 9, 10 to 14, and 15 or higher are considered to be *mild*, *moderate*, and *severe,* respectively.

The Parenting Sense of Competence (PSOC) [27] was used to assess parenting competence. The PSOC is a 17-item scale that assesses the domains of parental efficacy and satisfaction using a six-item Likert scale ranging from 1 (*strongly disagree*) to 6 (*strongly agree*), with higher scores indicating higher parenting competence.

The participants were also asked to rate the helpfulness of the texting platform in reminding them of the main ideas discussed in the sessions on a scale of 1 to 9 (1, not at all helpful; 9, most helpful). In addition, the frequency of replies to texts was also calculated. Additional measures were collected at T1, T2, and T3 that are not reported in this paper.

##### Qualitative

Following the completion of the intervention, the mothers participated in an online focus group to provide feedback regarding the intervention and changes for future interventions. The participants discussed their experience participating in the group, provided feedback on the content and structure of the group, and gave suggestions for future groups.

The focus group was conducted and recorded in Zoom and transcribed verbatim by the research team. The qualitative data analysis was guided by a thematic analysis, a useful method for examining the participants’ different perspectives and experiences, highlighting similarities and differences, and generating new insights [28,29]. We followed the six phases of a thematic analysis [28]. First, the research team familiarized themselves with the transcriptions and generated initial ideas. Second, the team came up with initial codes based on step 1. Third, the team searched for potential themes, gathering additional data from the transcript. Fourth, the team reviewed these themes, and fifth, they began defining and naming the various themes and subthemes. This process led to the formation of a codebook of broad themes and subtopics. Two members of the research team independently coded each transcript using the developed codebook. All four members of the research team came together to discuss coded excerpts, allowing the themes to emerge inductively from the transcript and reach saturation, and concluded with consensus agreement. Collectively, the team generated a set of quotes depicting these themes to produce a final report (i.e., in this manuscript and step 6 of thematic analysis).

## 3. Results

### 3.1. Quantitative

Table 3 shows the median domain scores across all three timepoints. Parenting sense of competence increased at each time point from 69.00 (IQR = 53.5–78.3) at T1 to 79.50 (IQR = 66.0–85.5) at T3. There was a slight increase in the median anxiety score at T2, likely due to one participant having a drastic increase in their score, but anxiety decreased from 5.0 (IQR = 0.0–13.3) at T1 to 4.0 (IQR = 1.0–7.3) at T3.

Figure 1 and Figure 2 show the participants’ total depression and anxiety scores by time point. For depression, most scores trended down over time; however, one participant had a sharp increase at T2, and another two participants had a slight increase in depressive symptoms from T2 to T3 (Figure 1). For anxiety, most scores trended down or remained stable over time; however, one participant had a sharp increase at T2 (Figure 2).

The participants attended a mean of 7.2 sessions (SD = 0.74), and all the participants remained in the study throughout the three time points.

#### Texting

Our research team sent a total of 301 text messages during the intervention. These comprised 14.2% skill reinforcement, 13.8% homework reminders, 15.0% mindfulness content, 21.3% self-monitoring activities, and 35.2% logistics texts. The participants replied most to the logistic texts (44.3%), followed by homework reminders (37.2%) and mood rating texts (21.9%). The participants also reported that the texts were very helpful (M = 7.7, SD = 1.5). Additionally, on average, they rated their moods as high (M = 6.9, SD = 1.8).

### 3.2. Qualitative

Five of the six participants attended the focus group. Overall, the results indicate that the participants felt positive about participating in the virtual group. Using a thematic analysis, four general themes emerged from the focus group. The first two focused on the process of being in the group, including increased social support and gaining a sense of empowerment. The third and fourth themes included the participants’ feedback on the content and structure of the intervention. The four themes are interconnected, as summarized and depicted in Figure 3.

#### 3.2.1. Increased Social Support

The first theme described the connections that mothers had as a result of participating in the virtual group. They enjoyed having a connection with other mothers. One mother stated, “*Being able to talk to people and support each other. And I like the breakout sessions where we could just kind of talk about whatever the things were and just be open.*” [participant 6]

Another subtheme arose in which the participants reported that they valued the information and advice that they received from other mothers. “*I liked that I could talk to other mothers and basically kind of vent and also learn different ways and techniques of how to help with raising my child. So it [group] really did help it at a time where I needed it.*” [participant 1]

#### 3.2.2. Empowerment and Free of Judgment

All the participants reported feeling “empowered” to share their personal experiences in the group. “*I felt more empowered as a mother, and you start to realize that you’re not the only person who went through certain stuff, so the advice that others give you can help you in your situation.*” [participant 3]. The mothers also appreciated having a safe space to share experiences in a “judgment-free zone”.

*Well, with the group I was able to express how I really felt. Like, when it came to being a single mother how everything… sometimes it can be hard. But with friends and family, they don’t really… they probably understand, but they just, most people think it’s just an excuse that you’re making to be able to complain. And they try to give unsolicited advice, with the group you don’t get that, you get people who listen. Instead of judging me, just listening*.[participant 3]

#### 3.2.3. Feedback on Content

The participants shared their thoughts about the content and information received during the group. Generally, the participants wanted the group to cover more topics about child development, felt that they were not fully engaged with certain group activities (e.g., guest speakers), and needed more time to connect with the material.

In reference to the reproductive psychiatrist (the guest speaker), a participant stated:

*I wasn’t too fond of, but it was just me personally at the time it didn’t really necessarily apply, but I feel like that might have been helpful to other people. So, it’s always good to have time to discuss whatever it is that we’re talking about, whatever the topic is. And it’s good to get information. But just also, just to you know see how maybe it’s applying to everybody and to have us to be able to talk about it*.[participant 6]

The participants appreciated having a physical copy of the workbook, something that they could review in their own time.

*I think the book is something I would look back at just for tips when things come up. But it was helpful just for finding out, or just having to think about certain things like, you know, like making a plan in advance for situations. So, it’s probably something to look back to just to remind myself*.[participant 5]

#### 3.2.4. Feedback on Structure

Most participants felt that the amount of time spent in each group session was appropriate. Some participants wished that the intervention was longer, up to 12 weeks, particularly among those who had younger infants and wanted the ongoing support from the group. “*I think it could have been more…You need a 12-week session, maybe.*” [participant 6]

Regarding the intervention’s format, most participants agreed that a virtual format is preferred. Some participants expressed that a hybrid format may have been more helpful for fostering social connections between participants and their children.

*I think being virtual was helpful. Even just being a new mother and different schedules of sleep and different things. It was good to just go online and start talking and see other people. I think it would be good, to maybe have some sessions, or maybe an evening session or something to be in person, so that you can connect to the people that you’re talking to, but the general virtual nature and yeah and just in person at the beginning, or at the end do something, or both*.[participant 6]

## 4. Discussion

The goals of this study were to describe an adaptation of the HealthySteps Moms Virtual Wellness Group for AA mothers with young children within a pediatric setting and to conduct and report results from a mixed-methods pilot study to examine the feasibility and acceptability of this adapted intervention. HealthySteps DC recognized an opportunity to develop and implement a necessary yet often neglected form of care for mothers during a crucial time in their lives and the lives of their children. The HealthySteps DC team adapted the MBC, an evidence-based preventive intervention for perinatal depression among low-resourced communities, to decrease the risk of PMADs and increase support for AA mothers navigating early motherhood during COVID-19—a time where Black birthing individuals had higher rates of perinatal depression and anxiety [16].

The results from the pilot study indicate that a virtual wellness group was both acceptable and feasible for the six AA mothers. The original intent of this group was to support mothers who were at risk for PMADs during COVID-19. However, the results of the quantitative surveys indicated that our sample had overall low depression and anxiety scores pre-intervention, which generally remained stable or decreased over time except for one participant. This participant experienced higher levels of anxiety and depression, which was associated with a significant health stressor experienced midway in the intervention, and the skills taught through the intervention were not enough to mitigate this stressor. Furthermore, the scores from our COVID-19 measure indicate that the participants did not experience a significant impact; this may have been due to the timing of when the intervention occurred—September to November 2021—which did not have a significant surge and the fact that the participants were then used to the pandemic, as it was 1.5 years into this period. In contrast, the participants reported an increase in parenting competence at T2 and T3 as a result of participating in the virtual group intervention. This was consistent with the qualitative data suggesting that the mothers benefited from participating in the virtual wellness group. They reported feeling supported by the other mothers in the group and felt “safe”, as evidenced by one participant’s description of the group as “a judgment-free zone.” It is possible that this support within a safe space contributed to the decline in depression and anxiety among the participants, which is a clinically significant finding. Additionally, the participants recommended a longer group (from 8 to possibly 12 sessions) with a hybrid format to foster optimal social connections.

In general, our quantitative results did not indicate decreased depression or anxiety among our group of mothers. This may have been due to the sample’s low depression and anxiety scores pre-intervention. Other studies that did demonstrate evidence of decreased depression with group interventions included participants with low-to-moderate and high risks of depression [30]. However, at baseline, our participants did not report high levels of depression or anxiety, which may explain our differing results. In addition, as stated previously, the group participants may have also received mental health support as a part of their involvement in the HealthySteps program in their clinic. Furthermore, other researchers have noted that AA mothers tend to underreport symptoms of PMADs due to a variety of structural and historical reasons, and current methods for detecting PMADs may be less effective or culturally relevant for this group [15].

Despite the potential barriers to the telehealth modality (i.e., lack of reliable WIFI connection, access to technology, privacy concerns, etc.), the mothers in this group participated regularly and consistently, as evidenced by the 100% retention rate. This may have been, in part, due to the culturally adapted intervention, low overall rates of perinatal depression and anxiety, and the use of texting, which the participants used and interacted with frequently during the intervention and contributed to the increased engagement and attendance in the virtual sessions. The bidirectional texting platforms embedded into interventions can be a useful and acceptable digital tool to reinforce wellness strategies, coordinate logistics, and increase participant engagement in maternal mental health interventions.

Overall, the results of the current study support the findings of others which indicate the importance of social support for addressing PMADs through virtual formats. While there do not appear to be any studies that examine the use of a virtual group for AA mothers during COVID-19, there are studies that have examined similar aspects to the current study with similar results. As has been demonstrated by others, this study found that social supports are a key protective factor against PPD [5]. Our mothers talked specifically about the importance of having the “no judgment zone” of the group, and depression levels decreased over time for most of the mothers in the group. The virtual format of the group was an innovation created out of necessity due to the COVID-19 pandemic. The mothers reported that the virtual format met their logistical needs and allowed them to participate more easily. This is consistent with findings from Ollivier et al. [31], who reported that many new mothers sought online support during the pandemic as a way to meet their mental health needs.

### Limitations

This study had several limitations. First, the biggest study limitation is the small sample size, which restricts the ability to determine if changes with respect to depression and anxiety were statistically significant. Nevertheless, the six mothers provided rich and valuable feedback on their experiences in participating in a virtual intervention during COVID-19; this small sample size was sufficient to explore a previously unexplored phenomenon (i.e., mental health experiences and engagement with virtual interventions in a specific group of African American mothers with young children during COVID-19) [32,33]. Second, logistical barriers (e.g., staff were not able to see patients in person to recruit; difficulty contacting participants via phone as some phones were disconnected) contributed to difficulties in recruitment [34]. However, we were able to recruit a small number, which was most successful when the recruitment conversations came from trusted providers. Third, we did not have a control group as this study was intended to provide mental health services during a time when such services were restricted within a pediatric setting. Finally, this study may have limited geographic (DC-based) and cultural (AA/Black population) generalizability.

## 5. Conclusions

In summary, our study indicates that a virtual wellness group was acceptable and feasible for AA mothers with young children during the pandemic, who had generally low depression and anxiety and increased parenting competence as a result of participating in the study. An examination of the past four years of the pandemic, its impact on AA families, and the resiliency of these same families highlights the strength and resolve that exists among AA families, which enables them to persevere and, in many cases, thrive. Murry et al. [35] have highlighted the importance of “cultural strength-based coping assets” within Black families, which include cultural legacies, family cohesion, racial identity, and kinship support. These assets may contribute to the strength and resiliency of this group of mothers.

When thinking about resilience in the context of AA parents and families, it is important to note that while it is true that historically African Americans have by necessity been resilient, this does not negate their need for support, advocacy, and intervention of all kinds and at many levels. The narrative of the “strong Black woman”, while rooted in the need for African American women to be strong, brings with it the risk of invisibility, often at times when it is most important for them to be seen and heard. Future research should focus on identifying resiliency factors among AA mothers and widening the scope of this intervention to include more participants to best understand the impact of the program on mothers and babies.

## Figures and Tables

**Figure 1 ijerph-21-00390-f001:**
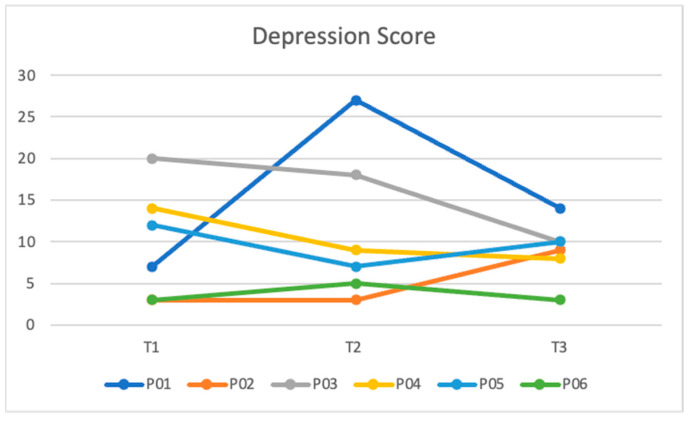
Total depression score by participant by survey timepoint.

**Figure 2 ijerph-21-00390-f002:**
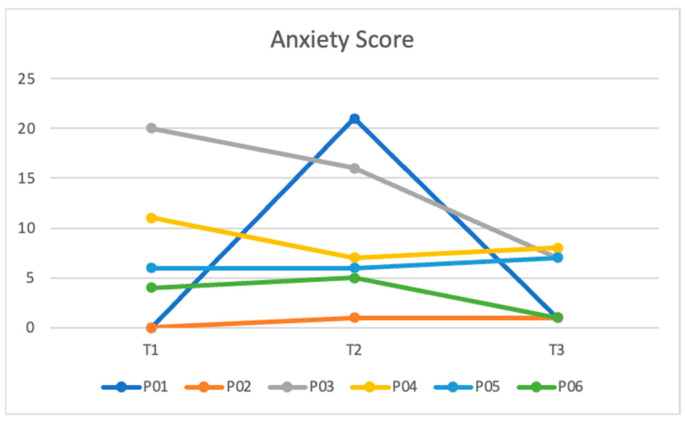
Total anxiety score by participant by survey timepoint.

**Figure 3 ijerph-21-00390-f003:**
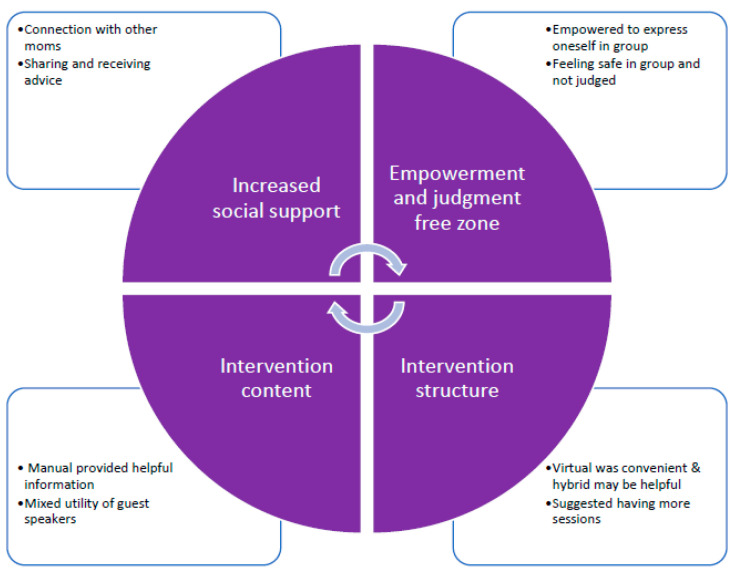
Themes from post-intervention focus group.

**Table 1 ijerph-21-00390-t001:** Content of the HealthySteps Moms Virtual Wellness Group.

Session	Topic	Activity
1	Introduction to the Group; Resources for Moms and Babies/Children	Welcome and introductions Review group guidelines Identify how stressors affect the mother–baby relationship Identify practical needs of the group Provide local resources How the group can help participants Introduce and practice mindfulness: awareness of breath Introduce Quick Mood Scale (weekly personal project) Introduce the use of texts ^1^
2	Being the Best Mother That I Can Be	Review Quick Mood Scale Mindfulness practice: loving kindness Thoughts about being a mother Identify helpful and harmful thoughts about motherhood Affirming identity as a Black woman and mother Assign Quick Mood Scale
3	Self-Preservation: Taking Care of Myself So I Can Take Care of My Baby	Review Quick Mood Scale Mindfulness practice: anchor breathing Reality management model and CBT Pleasant Activity List Relationship between behavior activation and mood Assign Quick Mood Scale
4	Mind Over Mood	Review Quick Mood Scale Mindfulness practice: body scan Define thoughts and how they affect mood Types and strategies to decrease harmful thought patterns Guest speaker: reproductive psychiatrist Assign Quick Mood Scale
5	Getting Support to Be the Best Mother That I Can Be	Review Quick Mood Scale Mindfulness practice: self compassion meditation Discuss how contact with others can influence mood Identify one’s social support network and types of support Assign Quick Mood Scale
6	How to Get My Needs Met	Review Quick Mood Scale Mindfulness practice: participant choice How to identify and communicate one’s needs Assign Quick Mood Scale
7	Parenting, Bonding, and Attachment	Review Quick Mood Scale Mindfulness practice: Guiding meditation for parents. Separate topic: Understanding Baby Cues Sleep hygiene and behavior management of toddlers Guest speaker: pediatrician or developmental psychologist Assign Quick Mood Scale
8	Putting It All Together	Review Quick Mood Scale Mindfulness practice: breathing meditation Review “toolkit”—identify ways to manage one’s stress, mood, and anxiety Review and create postpartum wellness plans Discuss the use of drugs and alcohol and their impact Review participants’ needs and next steps

^1^ Text reminders were provided in between sessions to reinforce previous session materials, including reminders to monitor mood and practice mindfulness.

**Table 2 ijerph-21-00390-t002:** Sample demographics (N = 6).

Demographics	% (n)
*Gender*	
Female	100.0 (6)
*Race*	
African American/Black	100 (6)
*Age*	29.3 (6.0)
*Marital Status*	
Single	83.3 (5)
Married	16.7 (1)
*Education*	
High school diploma or GED	33.3 (2)
Some college	0.0 (0)
Trade/vocational School	33.3 (2)
Associate’s degree	0.0 (0)
Bachelor’s degree	0.0 (0)
Graduate degree	33.3 (2)
*Number of Adults in Household*	1.7 (0.5)
*Number of Children*	1.9 (1.3)
*Number of Children in Household*	1.5 (0.83)
*Parenting*	
With a partner	66.7 (4)
By yourself	33.3 (2)
With the help of family/friends	0.0 (0)
*Annual Household Income*	
USD 0–10,000	50.0 (3)
USD 10,001–20,000	0 (0)
USD 20,001–30,000	0 (0)
USD 30,001–40,000	16.7 (1)
USD 40,001–50,000	0 (0)
More than USD 50,001	33.3 (2)
*Employment Status*	
Employed Full-Time	33.3 (2)
Employed Part-Time	16.7 (1)
Unemployed, looking for work	33.3 (2)
Unemployed, not looking for work	16.7 (1)
*Age of Youngest Child*	Range: 3 weeks to 2 years
*COVID Impact*	18.67 (6.12)

**Table 3 ijerph-21-00390-t003:** Quantitative results (N = 6).

Construct	T1 (Mdn, IQR)	T2 (Mdn, IQR)	T3 (Mdn, IQR)
Parenting Sense of Competence	69.00 (53.5–78.3)	76.50 (67.3–82.0)	79.50 (66.0–85.5)
Depression	9.50 (3.0–15.5)	8.00 (4.5–20.3)	9.50 (6.8–11.0)
Anxiety	5.00 (0.0–13.3)	6.50 (4.0–17.3)	4.00 (1.0–7.3)

## Data Availability

The data that support the findings in this study are available from the corresponding author upon reasonable request.

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
