# Peer review of "“A Judgment-Free Zone”: Adaptation and Pilot Study of a Virtual Wellness Group for African American Mothers with Young Children"

_ijerph, 2024, doi:10.3390/ijerph21040390_

Round 1

Reviewer 1 Report

Comments and Suggestions for Authors

1- It could be interesting to go in deep the sentences "black people experienced more symptoms and severity of PPD" in the introduction. 

2- The COVID-19 pandemic increased the risk of PPD but before the pandemia the results weren't different. Maybe, you could add a reflection in this topic (see Stephanie V Hall - Racial Disparities in Diagnosis of Postpartum Mood and Anxiety Disorders Among Symptomatic Medicaid Enrollees, 2012-2015)

3- Your study is interesting and it could be improved by noting the differences with other studies like the study of Keilondria Robertson - Black with 'Baby Blues': A Systematic Scoping Review of Programs to Address Postpartum Depression in African American Women

4- It could be useful to add if the 6 women were treated with antidepressants or anxiolytic medicines and a general sentence about the reproductive psychiatrist consultation

Reviewer 2 Report

Comments and Suggestions for Authors

This paper presents needed evidence regarding the effectiveness of virtual parenting programs.  There are some suggestions for improvement:

1)  The section at the beginning of the Introduction that addresses the challenges of the pandemic should include more literature discussing the challenges of the pandemic specifically for African American families.

2)  In Section 2.1.1., how was the Mothers and Babies Course specifically adapted for this intervention?

3)  In Section 2.2.3., the two pediatric primary care sites need more description - where are they located?  What size?

4)  In the demographics section, there should be a statistic under Marital Status that gives the percentage of mothers who cohabit with their partners while they are unmarried.

5)  In the Measures section, example items for each subscale and coefficient alphas are needed for all measures.

6)  In the Qualitative section, it is unclear how coding was done.  How did the inductive thematic coding process work?  Were there codes that were grouped into themes, such as in thematic analysis?  There also needed to be reliability information given between coders.

7)  In Section 3.1, Cohen's d effect sizes should have been given to provide a measure of the magnitude of all effects.

8)  Lines 241-242 were stated in the Methods section and do not need to be repeated.

9)  In the Texting section, it is unclear how it was decided to send which particular text content and when.  This information should be given in the description of the program.

10) The Discussion mentions that participants had low levels of depression and anxiety.  The authors should suggest future studies with clinical samples.

11) In line 368, what were the "logistical barriers"?

12)  The limitation of geographic and cultural generalizability to other ethnic groups and locations should be mentioned.

Reviewer 3 Report

Comments and Suggestions for Authors

universal services including #2 (line 124) - is confusing

Line 127-128 - missing a word

A style suggestion is to put a Limitations Heading in the paper instead of including it in the discussion.  You might want to build this section out just a bit more as you have mentioned some limitations such as the level of depression and anxiety earlier in the paper.

Other than that - well done and very interesting to read.

Thank you

Reviewer 4 Report

Comments and Suggestions for Authors

The study presents an adapted virtual wellness group, "HealthySteps Mom's Virtual Wellness Group," designed for African American mothers with young children, aiming to address the unique mental health challenges exacerbated by the COVID-19 pandemic. Given the limited sample of 6, I would strongly suggest author to provide a clear rationale for limited sample. The prime focus of the paper should be in qualitative as quantitative data would not yield any significant statistical value. 

The authors can contextualise how COVID-19 has influenced the rates of PMADs. For instance, elaborating on the factors such as social interactions, social support, isolation, and changes in perinatal care that contribute to the increased incidence.

Methods:

Provide specific dates or durations for Phase 1 activities, such as when the HealthySteps Moms Virtual Wellness Group was conducted, to give readers a clear timeline of the intervention development.

Elaborate on the strategies used for participant recruitment, including any incentives offered, and discuss the role of Family Service Coordinators (FSCs) in the recruitment process.

Include more information about the qualifications and expertise of the clinicians who facilitated the group sessions in Phase 1. Specify their background in maternal mental health and early childhood expertise.

Provide more specifics on the nature and content of the mindfulness exercises conducted at the beginning of each session. This could include the type of exercises, duration, and their specific relevance to the session topics.

Clarify the specific topics covered by the guest speaker (reproductive psychiatrist) in session 4 and the paediatrician or developmental psychologist in session 7. Specify how these topics were integrated into the overall intervention.

Provide a more detailed explanation of the qualitative analysis process, including how themes were identified, the coding procedure, and the criteria for consensus agreement among the research team.

The authors must provide a clear rationale of limited sample of 6, was data saturation attained in the qualitative results. 
Given the sample being 6 individuals, quantitative analysis yields limited value. Author can actually make the descriptive very brief and move two graphs to supplementary material.

More details on qualitative theme are advised.

Comments on the Quality of English Language

None

Round 2

Reviewer 1 Report

Comments and Suggestions for Authors

No other suggestions

Author Response

Reviewer 1 had "No other suggestions." 

Response: Thank you for your review. 

Reviewer 4 Report

Comments and Suggestions for Authors

Dear Authors,

Given the limited sample size of 6, I strongly suggest providing a clear rationale for this choice and justifying the continuum of sizes that can be justified by researchers. Thank you for shifting the focus of the paper to qualitative aspects. I still recommend suggesting why a sample size of 6 is adequate, especially for critical topics in mental health of vulnerable group. Here are some resources that may guide the author in justifying the narrative:

https://www.mdpi.com/2227-9032/11/19/2665

https://journals.sagepub.com/doi/abs/10.1177/1049732315617444

https://www.emerald.com/insight/content/doi/10.1108/qmr-06-2016-0053/full/html

https://digitalcommons.tacoma.uw.edu/socialwork_pub/500/

Comments on the Quality of English Language

None
